# Impact of outer width of the metacarpal diaphysis on the identification of low bone mass in children

Samantha Hertz[1], Finnegan Klein[1], Todd L. Bredbenner[2], Miranda Cosman[1], Karl J. Jepsen[1]*

1 University of Michigan, Ann Arbor, Michigan, United States of America, 2 University of Colorado Colorado Springs, Colorado Springs, Colorado United States of America

* kjepsen@umich.edu

## Abstract

Developing a strong skeleton during growth is critical for minimizing fractures later in life. Prior work showed that bone mass varied with external bone size, a measure of the outer bone width. We tested how this association affected the identification of children with low bone mass. Radiographs of the nondominant hand of 45 White females and 54 White males, all ~8 years old, were assessed and second metacarpal length (Le) and the midshaft outer and inner widths were measured at the 40, 50, and 60% midshaft sites. The average total area (Tt.Ar), a measure of the area enclosed by the periosteal surface, and cortical area (Ct.Ar), a measure of bone mass, were calculated assuming a circular cross-section. Individuals were sorted into tertiles using robustness (Tt.Ar/Le). Z-scores were calculated for Ct.Ar first using the cohort mean and standard deviation and second using each robustness tertile mean and standard deviation. Females and males with Z-scores in the lower 33% range were identified for the group-average and tertile-specific average comparisons. Agreement between the two reference group approaches was determined using Cohen's kappa statistic for each sex. The percentage of individuals identified with low Ct.Ar depended on whether Z-scores were compared to the group average or tertile-specific averages. When compared to the group average, 67% of females and 56% of males identified with lower Ct.Ar were from the narrowest tertile, whereas 0% of females and 22% of males were from the widest tertile. For females and males, Cohen's kappa coefficient showed almost perfect agreement for the intermediate tertile (kappa coefficient > 0.84), but agreement was only poor to moderate (kappa coefficient < 0.53) for the narrowest and widest tertiles. Our findings provide evidence that external bone size may be a source of bias when assessing bone mass in females and males. Our analysis suggests that comparing morphological parameters from a child's metacarpal to their structural peer (i.e., reference group with a similar external size) may allow for more precision in diagnosing low bone mass and potentially lower bone strength.

**Data availability statement:** The measured and calculated metacarpal data used in this study are available in a publicly accessible resource. The doi for this data is: https://doi.org/10.7302/47ej-xb63 Hand X-rays are publicly available through https://case.edu/dental/departments-programs/bolton-brush-growth-study-center.

**Funding:** National Institute of Arthritis and Musculoskeletal and Skin Diseases of the National Institutes of Health (AR069620, AR068452;AR064244;AR082325).

**Competing interests:** The authors have declared that no competing interests exist.

## Introduction

Developing a strong skeleton during growth is critical for minimizing pediatric fractures [1] and fracture risk later in life [2]. Bone strength is not routinely assessed in healthy, ambulatory children and typically only considered in cases of concerning bone age evaluations, disease conditions, dysplasias, multiple fractures [3], or idiopathic juvenile osteoporosis [4]. The current approach to assessing bone strength is tailored for rare diseases and related conditions but is neither beneficial for the general population nor effective in establishing comparative norms. Given the trend toward increased fracture incidence in children worldwide [5–7], it is important that bone health assessments allow for an understanding of how factors such as exercise [8–12], nutrition [13–16], and environmental exposures [17] affect bone strength development and fracture risk [18]. To accomplish this goal, it is critical that tools used to monitor bone strength in children estimate strength from measures of bone mass and structure consistently among individuals and with minimal biases [19].

A review of morphological measures derived from hand radiographs revealed that many widely used measures correlated with experimentally determined strength but with a sex-specific discrepancy [20]. Another potential source of bias which may not be immediately obvious beyond sex and race/ethnicity is the variation in bone structure itself [20]. One structural variable that is important from a strength standpoint is external bone size. External bone size represents any measure describing the outer size of the bone, including outer width and total cross-sectional area. Because bending strength is related to the third power of outer bone width, small differences in external bone size can lead to large changes in strength. External bone size varies within female and male populations, ranging from narrow to wide phenotypes, even after adjusting for body stature [21,22]. Moreover, the variation in external bone size is associated with coordinated changes in multiple traits (e.g., marrow area, cortical thickness, porosity), resulting in narrower long bones having a different combination of traits to establish strength compared to wider long bones (Fig 1). Narrower bones have proportionally thicker cortices but lower Ct.Ar (i.e., mass) compared to wider bones [22,23]. Despite this coordinate adjustment of traits, narrower long bones are 1.3–2.8 times less strong compared to wider bones [24]. Thus, within any population there exists layers of variation that should be considered when attempting to be precise in bone health assessment.

To test for low bone mass, a child's bone mass is generally compared to a reference group, which is often the population average for their sex and age [25]. As noted above, the variation in external bone size and associated trait combinations has the potential to be a source of bias if not taken into consideration. We propose that consideration of external bone size and the associated trait combinations may allow for more precision in assessing low bone mass in children. The current study builds on our prior work which established different external size dependent longitudinal trajectories for bone traits during growth [23,24]. Our prior work showed that the morphological traits of narrow bones grow along a different trajectory compared to wide [26,27]. In the current study, we tested whether assessing low Ct.Ar can be improved by taking the variation in external size into consideration. For

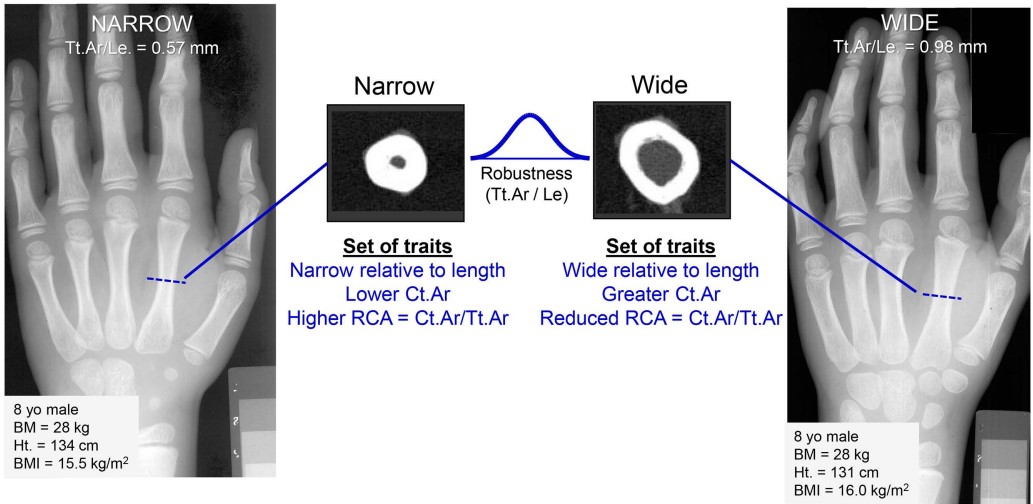

**Fig 1. Schematic of hand radiographs of two 8-year-old males enrolled in the Bolton-Brush collection matched for body stature.** The radiographs illustrate the morphological differences between narrow and wide metacarpals and the associated sets of traits. The metacarpal cross-sections were taken from adult metacarpals using a peripheral quantitative computed tomography system and were not matched to the hand radiographs but are included to illustrate the differences in cross-sectional structure between the two external bone size phenotypes.

the metacarpal, Ct.Ar is a direct measure of the amount of bone (i.e., bone mass) for the diaphysis and is correlated with whole bone strength without a sex-specific bias [20]. Given that Ct.Ar varies with external bone size, we tested the hypothesis that a comparison of an individual's Ct.Ar to the group average will result in an over-diagnosing of low Ct.Ar in narrower bones and under-diagnosing of low Ct.Ar in wider bones compared to using a reference group comprised of subgroups with similar external sizes.

## Materials and methods

### Sample population and data collection

Our analysis utilized data from the Bolton-Brush study, a longitudinal dataset which includes posteroanterior (PA) radiographs of the nondominant hand of White children from Cleveland, Ohio, collected around 1930. The majority of the radiographs examined were of the left hand, which is consistent with the general statistic that approximately 90% of individuals are right hand dominant. No IRB approval was needed because the archived data is deidentified and publicly available. Digitized hand radiographs were purchased from the Broadbent-Bolton Collection at the Bolton-Brush Growth Study Center in 2007 and represent a randomly selected subgroup of those enrolled in this study. The authors did not have access to information that could identify individual participants at any time. Radiographic images were the only available technology in 1929 to provide a noninvasive assessment of the human skeleton. The Bolton-Brush collection was used because it was available to the authors to test the primary hypothesis as a proof-of-concept. The current study used this previously reported data but combined the birth – 8 year old [27] and 8–18 year old [26] cohorts in a new way to provide the power needed to test the current hypothesis. The current datasets included 45 females and 54 males who were approximately eight years old. The female and male datasets allowed us to test for an effect size as low as 0.45 when comparing the tertiles (3 groups) with 80% power and a significance level of 0.05. This study was powered to test for significant differences in Ct.Ar among tertiles, which showed an average effect size of 1 for females and males when analyzed separately. Data included body weight and height, when available, and all morphological measures quantified from hand radiographs (see below). Body Mass Index (BMI) was calculated as body weight (kilograms) divided by height squared (meters$^2$).

## Bone structural measures

All methods were the same as those described in prior studies [26,27]. Briefly, LabVIEW Vision Builder software (National Instruments, Inc; Austin, TX USA) was used to quantify the diaphyseal structure of the second metacarpal. Metacarpal length (Le) was measured manually from the proximal end of the metacarpal to the proximal end of the distal growth plate. This length measure was used because the growth plate was visible on the radiographs and provided a defined anatomical reference point compared to the variable surfaces provided by the developing secondary center of ossification. Linear dimensions of the outer (periosteal, D) and inner (endosteal, d) surfaces were measured manually at 10% increments along the diaphysis. Manual point-to-point measurements refer to the identification of the edges of the outer and inner surfaces of the diaphysis by an individual and not by an edge detection algorithm. A repeatability study was conducted at the time the measurements were taken using a random collection of five hand radiographs with 10 repeat measurements for each of the outer and inner widths. Manual point-to-point measurements of outer and inner bone widths had an average coefficient of variation of 1.61%. The midshaft structural data reported herein are the average of measures taken at 40, 50, and 60% along the length of the metacarpal. Total cross-sectional area (Tt.Ar = $\pi$ (D/2)$^2$), marrow area (Ma. Ar = $\pi$ (d/2)$^2$), and cortical area (Ct.Ar = Tt.Ar − Ma.Ar) were calculated assuming the shape of the diaphyseal cross-section was a concentric circular cylinder. Assuming the metacarpal cross-section resembles a concentric cylinder is consistent with prior work and is appropriate given that standards for out of plane shape differences have not been established [20]. Robustness was calculated as the ratio of Tt.Ar to length (Tt.Ar/Le). Robustness is a measure of external bone size that is adjusted for the variation in body stature. Normalizing Tt.Ar by bone length takes body size into consideration, allowing us to study the variation in external bone size while minimizing the effects of body size. Ct.Ar is a measure of the amount of bone and is directly proportional to bone mass (bone mass = bone volume x density = Ct.Ar x h x density, where h = height of the region of interest). Ct.Ar was used as a measure of bone mass rather than other parameters used in radiogrammetry such as the metacarpal index (MCI = cortical thickness normalized by total width) [28], because the association between parameters like MCI and whole bone strength showed a significant difference between females and males [20] and our goal was to minimize bias. Relative cortical area (RCA) was calculated as Ct.Ar/ Tt.Ar.

## Analytical methods

We tested two hypotheses in the current study with females and males being analyzed separately. First, we tested whether Ct.Ar differed among subgroups sorted based on external bone size. Second, we tested whether identifying individuals with lower bone mass would be improved when comparing Ct.Ar of individuals to the average of their external size subgroups versus comparing Ct.Ar of individuals to the overall cohort average. Although morphological traits like the moments of inertia tend to correlate with the resistance to bending and torsional loads of many long bones, our direct mechanical testing of the metacarpal revealed that Ct.Ar was the best morphological trait that correlated with strength and without a sex-specific difference in the strength-Ct.Ar regressions [20]. Thus, Ct.Ar was used herein as a dual indicator of bone mass and a proxy for bone strength. The metacarpal is a short beam, and the outcome of these direct mechanical tests suggested the applied bending loads induced shear forces. As such, the use of Ct.Ar as a morphological measure that correlates with strength is consistent with engineering principles.

As in our prior studies [26,27], participants were sorted into tertiles using robustness (Tt.Ar/Le) for females and males separately. Normality was assessed using the Kolmogorov-Smirnov test. Averages and standard deviations were calculated for demographic and morphological measures using all data (group average approach) and for each robustness tertile (tertile-specific average approach). Significant differences in trait values between the tertiles were determined using a one-way ANOVA and Tukey post-hoc test. The schematics of the two approaches used to establish reference groups are shown in Fig 2.

Variation in Ct.Ar was determined by converting Ct.Ar values to Z-scores. Z-scores for Ct.Ar were calculated for females and males separately, first by comparing individuals to the mean and standard deviation of the overall cohort

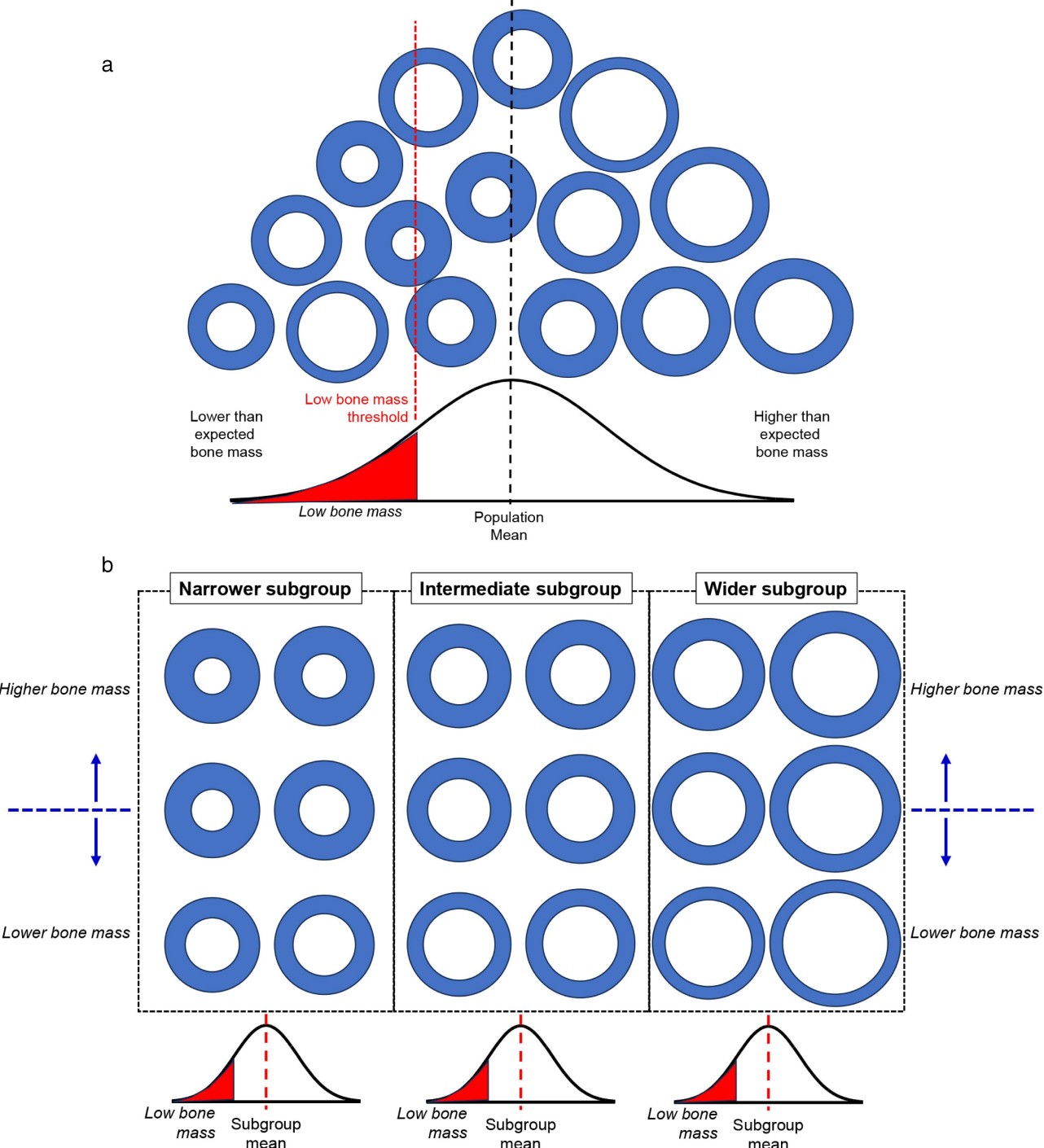

**Fig 2. (A) Schematic illustrating the variation in bone mass among idealized cross-sections of metacarpals.** The cross-sections were organized based on Ct.Ar (lowest to highest), which results in the overlapping of external sizes. Ct.Ar is a direct measure of bone mass for the metacarpal and is normally distributed among individuals. A low bone mass threshold is shown along with the type of bone structures with lower-than-expected bone mass for the population. **(B)** A similar distribution of the idealized metacarpal cross-sections shown in Figure 2a is reorganized in this schematic to illustrate how cross-sections can be sorted **first** based on external bone size (horizontal axis) and **second** by Ct.Ar (vertical axis). The external bone size phenotypes ranged from narrow to wide. Within each robustness tertile exist individuals whose Ct.Ar varies from lower to higher than expected for their external bone size. This tiered organization process reveals how individuals should be compared to their external size peers to determine if they have lower than expected bone mass.

(group average approach) and second by comparing individuals to the mean and standard deviation of their robustness tertile (tertile-specific average approach). Females and males having lower Ct.Ar were identified as having Z-scores in the lower 33% range for both the group-average and tertile-specific average comparisons. To ensure both comparison methods had the potential for overall agreement, Z-score thresholds were varied among the tertiles, so the total number of individuals identified as having low Ct.Ar was the same for both methods. Cohen's kappa coefficient was used to test for agreement between the two reference group methods. A sensitivity analysis was conducted to determine if agreement between the two methods depended on the lower Ct.Ar threshold. For this, Cohen's kappa coefficient was determined when the percentage of individuals with lower Ct.Ar was increased to 50% and decreased to 25%. Finally, to test whether the process of sorting individuals into tertiles affected the agreement between the two methods, females and males were randomly sorted into tertiles separately. The same methods described above for calculating Z-scores (i.e., comparing individual values to the overall cohort versus individual tertiles) and Cohen's kappa coefficient were applied to the randomly sorted children. Only the 33% lower Ct.Ar threshold was examined for the randomization analysis. For clarity, the two sorting methods were designated as 'robustness tertiles' for the analysis that sorted individuals based on robustness and 'randomized tertiles' for the analysis that sorted individuals randomly into tertiles.

## Results

The average anthropometric and metacarpal structural measures for the 8-year-old females and males are shown in Table 1. Weight (Females: $F(2)=0.16$, $p=0.849$; Males: $F(2)=1.00$, $p=0.376$; ANOVA), height (Females: $F(2)=1.51$, $p=0.234$; Males: $F(2)=0.36$, $p=0.698$; ANOVA), and metacarpal length (Females: $F(2)=0.40$, $p=0.673$; Males: $F(2)=0.002$, $p=0.998$; ANOVA) did not differ among the robustness tertiles for either sex (Table 2), indicating that sorting females and males based on second metacarpal robustness reflected differences in outer bone size but not body stature or bone length. Robustness, Tt.Ar, and Ma.Ar differed among the tertiles, as expected. Relative cortical area (RCA) differed among the robustness tertiles for males but not females. The differences in Ct.Ar values among the robustness tertiles are illustrated in Fig 3 with bar graphs shown for the entire female and male cohorts (Fig 3A,D), those sorted into robustness tertiles (Fig 3B,E), and those sorted into randomized tertiles (Fig 3C,F). Ct.Ar differed significantly among the robustness tertiles for both sexes with the narrowest tertile showing lower Ct.Ar compared to the widest tertile ($p<0.05$, Tukey post hoc test) for females ($F(2)=31.2$, $p<0.0001$, ANOVA) and males ($F(2)=10.3$, $p=0.0002$, ANOVA). Sorting females and

**Table 1. Demographic, anthropometric, and metacarpal structural measures of the 8-year-old females and males included in the analysis. Data are shown as mean (standard deviation). The midshaft structural data are the average of the 40, 50 and 60% sites.**

| Variable | Female | Male |
|---|---|---|
| Age (yr) | 8.1 (0.2) | 8.1 (0.2) |
| Weight (kg) | 26.8 (5.2) | 28.2 (3.3) |
| Height (m) | 1.3 (0.1) | 1.3 (0.1) |
| BMI (kg/m^2) | 16.2 (2.3) | 16.5 (1.3) |
| Robustness (Tt.Ar/Le) | 0.65 (0.1) | 0.75 (0.1) |
| Length (mm) | 42.4 (2.6) | 42.7 (3.0) |
| Outer Diameter (mm) | 5.9 (0.5) | 6.4 (0.6) |
| Inner Diameter (mm) | 2.8 (0.6) | 3.3 (0.6) |
| Ma.Ar (mm) | 6.6 (2.6) | 9.1 (3.5) |
| Tt.Ar (mm) | 27.7 (4.8) | 31.9 (5.5) |
| Ct.Th | 1.5 (0.3) | 1.5 (0.2) |
| RCA | 0.77 (0.08) | 0.72 (0.08) |

**Table 2. Comparison of anthropometric and metacarpal structure for 8-year-old 2a) females and 2b) males sorted into robustness tertiles. The midshaft structural data are the average of the 40, 50 and 60% sites. Data are shown as mean (standard deviation). Differences among groups were determined using a one-way ANOVA with the Tukey post hoc test. The F-statistic and degrees of freedom (dof) are included. Tertiles with different superscript letters indicate a significant difference (p<0.05).**

**2a. Female tertiles**

| Parameter | Robustness Tertile | | | ANOVA |
| --- | --- | --- | --- | --- |
| | Narrow | Intermediate | Wide | |
| Weight (kg) | 26.5 (7.1) | 27.5 (5.2) | 26.4 (2.4) | $F(2)=0.16$, $p=0.849$ |
| Height (m) | 1.3 (0.5) | 1.3 (0.1) | 1.3 (0.04) | $F(2)=1.51$, $p=0.234$ |
| BMI (kg/m²) | 16.4 (3.5) | 16.2 (1.8) | 15.9 (1.2) | $F(2)=0.15$, $p=0.863$ |
| Robustness (Tt.Ar/Le) | 0.54 (0.04)[a] | 0.65 (0.02)[b] | 0.77 (.05)[c] | $F(2)=133.9$, **$p<0.0001$** |
| Length (mm) | 42.3 (2.6) | 42.0 (3.0) | 42.8 (2.2) | $F(2)=0.40$, $p=0.673$ |
| Outer Dia. (mm) | 5.4 (0.2)[a] | 5.9 (0.3)[b] | 6.5 (0.3)[c] | $F(2)=63.9$, **$p<0.0001$** |
| Inner Dia. (mm) | 2.4 (0.5)[a] | 3.0 (0.1)[b,c] | 3.2 (0.5)[c] | $F(2)=9.0$, **$p=0.001$** |
| Tt.Ar (mm²) | 22.8 (2.0)[a] | 27.4 (2.5)[b] | 32.9 (2.9)[c] | $F(2)=61.9$, **$p<0.0001$** |
| Ma.Ar (mm²) | 4.7 (2.0)[a] | 7.1 (2.3)[b,c] | 8.0 (2.6)[c] | $F(2)=8.3$, **$p=0.001$** |
| RCA | 0.80 (0.08) | 0.74 (0.09) | 0.76 (0.07) | $F(2)=2.2$, $p=0.128$ |
| Ct.Th (mm) | 1.50 (0.23) | 1.47 (0.29) | 1.66 (0.21) | $F(2)=2.6$, $p=0.084$ |

**2b. Male tertiles**

| Parameter | Robustness Tertile | | | ANOVA |
| --- | --- | --- | --- | --- |
| | Narrow | Intermediate | Wide | |
| Weight (kg) | 27.3 (3.4) | 28.4 (3.2) | 28.8 (3.4) | $F(2)=1.00$, $p=0.376$ |
| Height (m) | 1.3 (0.05) | 1.3 (0.1) | 1.3 (0.04) | $F(2)=0.36$, $p=0.698$ |
| BMI (kg/m²) | 16.1 (1.6) | 16.6 (0.8) | 16.6 (1.5) | $F(2)=0.93$, $p=0.400$ |
| Robustness (Tt.Ar/Le) | 0.62 (0.05)[a] | 0.76 (0.03)[b] | 0.87 (0.07)[c] | $F(2)=110.9$, **$p<0.0001$** |
| Length (mm) | 42.7 (3.0) | 42.7 (2.8) | 42.7 (3.3) | $F(2)=0.002$, $p=0.998$ |
| Outer Dia. (mm) | 5.8 (0.4)[a] | 6.4 (0.2)[b] | 6.9 (0.4)[c] | $F(2)=51.1$, **$p<0.0001$** |
| Inner Dia. (mm) | 2.8 (0.5)[a] | 3.3 (0.4)[b] | 3.9 (0.5)[c] | $F(2)=25.0$, **$p<0.0001$** |
| Tt.Ar (mm²) | 26.4 (3.1)[a] | 32.2 (2.0)[b] | 37.2 (4.3)[c] | $F(2)=49.0$, **$p<0.0001$** |
| Ma.Ar (mm²) | 6.3 (1.9)[a] | 8.8 (2.2)[b] | 12.2 (3.3)[c] | $F(2)=23.7$, **$p<0.0001$** |
| RCA | 0.76 (0.07)[a] | 0.73 (0.07)[a,b] | 0.67 (0.08)[a] | $F(2)=6.84$, **$p=0.002$** |
| Ct.Th (mm) | 1.49 (0.19) | 1.54 (0.25) | 1.45 (0.27) | $F(2)=0.62$, $p=0.545$ |

a, b, c – indicate the outcome of the Tukey post-hoc test. Tertiles with different letters indicate p<0.05.

males into the randomized tertiles resulted in no differences in Ct.Ar values among the tertiles for females ($F(2)=1.4$, $p=0.248$, ANOVA) and males ($F(2)=0.7$, $p=0.504$, ANOVA), as expected.

The percentage of individuals identified in the lower 33% of Ct.Ar values depended on whether the Z-scores were compared to the group average or tertile-specific averages (Table 3). For females, the Z-score thresholds were −0.51 and −0.40 so 33% of individuals were identified with low bone mass when compared to the group average and tertile-specific average, respectively. For males, the Z-score thresholds were −0.70 and −0.36 so 33% of individuals were identified with low bone mass when compared to the group average and tertile-specific average, respectively. When individuals were compared to the group average, 67% of females and 56% of males identified as having lower Ct.Ar were from the narrowest tertile, whereas 0% of females and 22% of males identified with lower Ct.Ar were from the widest tertile. When comparing individuals to their tertile-specific average, the percentages of females and males identified with lower Ct.Ar were more evenly distributed. Cohen's kappa coefficient showed almost perfect agreement for the intermediate tertile (kappa coefficient >0.81), but the agreement was only poor (kappa coefficient <0.20), fair (kappa coefficient between 0.21–0.40),

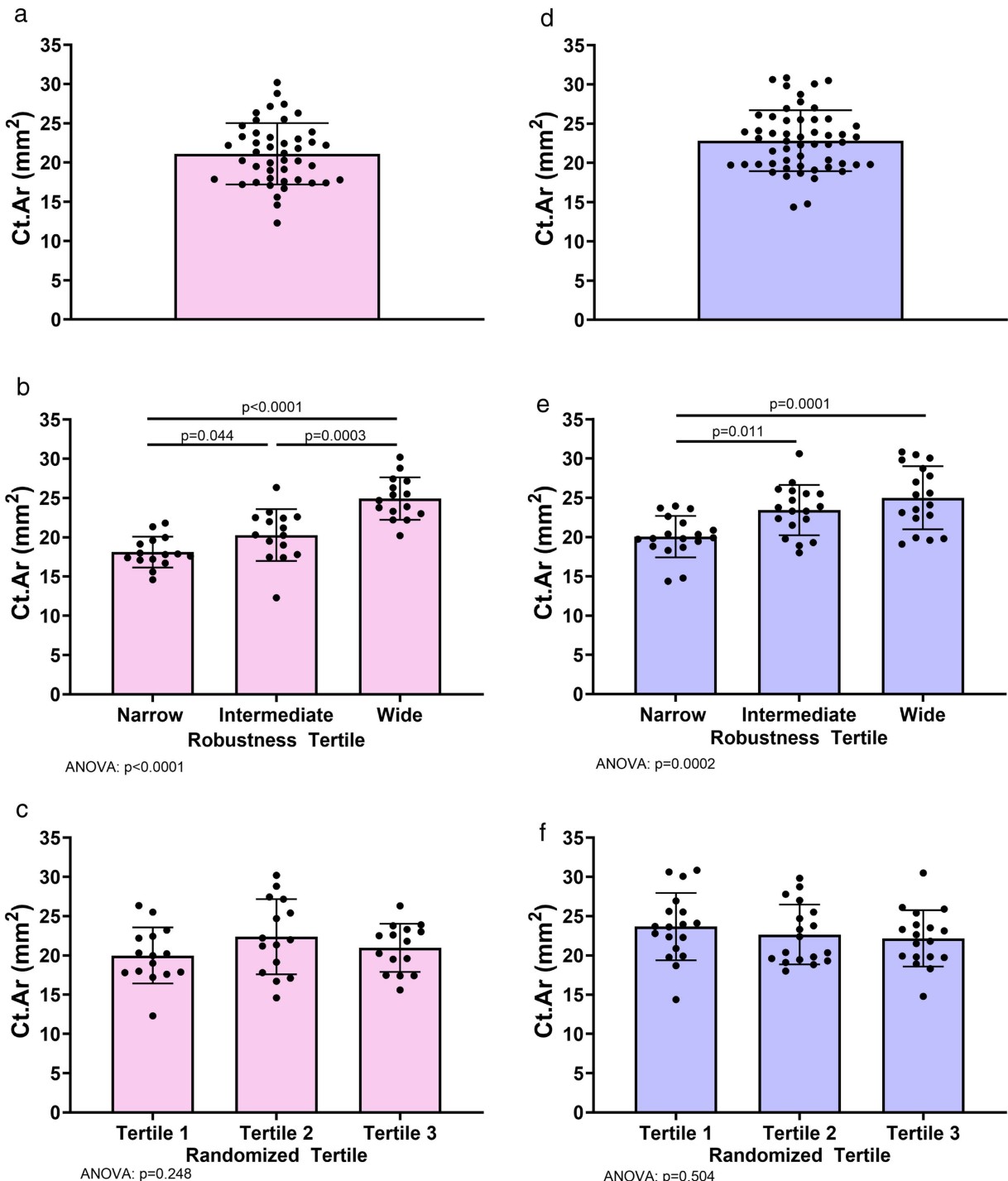

**Fig 3. Second metacarpal midshaft Ct.Ar values for 8-year-old females (A-C) and boys (D-F).** Data are presented as group averages **(A,D)**, averages after sorting individuals into robustness-tertiles based on metacarpal robustness (Tt.Ar/Le) **(B,E)**, and averages after randomly sorting individuals into randomized-tertiles **(C,F)**. Error bars represent standard deviations. The midshaft structural data are the average of the 40, 50 and 60% sites.

**Table 3. Percentage of individuals identified as having lower Ct.Ar when using group averages compared to sorting individuals into robustness tertiles for three threshold levels of lower Ct.Ar. The midshaft structural data are the average of the 40, 50 and 60% sites. Data are shown as percentage (n, number of children) for each robustness tertile. The Chi-square statistic is shown along with the total sample number (N) and degrees of freedom (dof).**

**3a. Females**

| Robustness Tertile | 50% threshold | | 33% threshold | | 25% threshold | | Randomized data (33% threshold) | |
|---|---|---|---|---|---|---|---|---|
| | Group | Tertile | Group | Tertile | Group | Tertile | Group | Tertile |
| Narrow | 59(13) | 41%(9) | 67%(10) | 33%(5) | 67%(8) | 25%(3) | 47%(7) | 40%(6) |
| Intermediate | 36%(8) | 27%(6) | 33%(5) | 27%(4) | 33%(4) | 33%(4) | 27%(4) | 33%(5) |
| Wide | 5%(1) | 32%(7) | 0%(0) | 40%(6) | 0%(0) | 42%(5) | 27%(4) | 27%(4) |
| *Chi-Square (N, dof)* | *0.007(45, 2)* | *0.727(45, 2)* | *0.007(45, 2)* | *0.819(45, 2)* | *0.018(45, 2)* | *0.789(45, 2)* | *0.549(45, 2)* | *0.819(45, 2)* |

**3b. Males**

| Robustness Tertile | 50% threshold | | 33% threshold | | 25% threshold | | Randomized data (33% threshold) | |
|---|---|---|---|---|---|---|---|---|
| | Group | Tertile | Group | Tertile | Group | Tertile | Group | Tertile |
| Narrow | 56%(15) | 37%(10) | 56%(10) | 28%(5) | 54%(7) | 23%(3) | 22%(4) | 28%(5) |
| Intermediate | 26%(7) | 30%(8) | 22%(4) | 28%(5) | 31%(4) | 38%(5) | 39%(7) | 39%(7) |
| Wide | 19%(5) | 33%(9) | 22%(4) | 44%(8) | 15%(2) | 38%(5) | 39%(7) | 33%(6) |
| *Chi-Square (N, dof)* | *0.045(54, 2)* | *0.895(54, 2)* | *0.135(54, 2)* | *0.607(54, 2)* | *0.232(54, 2)* | *0.735(54, 2)* | *0.606(54, 2)* | *0.846(54, 2)* |

or moderate (kappa coefficient between 0.41–0.60) for the narrowest and widest tertiles. Agreement between the two methods was consistent when the threshold for lower Ct.Ar was increased to 50% or decreased to 25%. Randomly sorting individuals into tertiles (randomized tertiles) showed almost perfect agreement between the group average Ct.Ar and tertile-specific average methods, confirming the sorting process itself did not artificially create the tertile-specific differences shown for the robustness tertiles.

## Discussion

Hand radiographs from eight-year-old females and males were examined and analyzed separately as a proof-of-concept to test whether metacarpal external size affected the identification of children with lower Ct.Ar. The ranges in external bone sizes reported in this study, which were consistent with prior work in adult bones [20], likely reflect influences of genetic [29–31] and environmental [32] factors. The primary outcome measure was metacarpal Ct.Ar which provides a measure of bone mass and a proxy for whole bone strength that lacks a sex-specific bias [20]. Two approaches to establishing a reference population were compared, one using overall group averages and the second using averages from the robustness tertile to which each child was sorted. Sorting females and males into sex-specific bone robustness tertiles confirmed Ct.Ar was lower in the narrowest tertile than the widest tertile, as reported previously [26,27]. Individuals identified in the lower 33% of Ct.Ar values differed depending on whether the criteria for lower Ct.Ar used group averages or tertile-specific averages. Using group averages, most individuals identified with low Ct.Ar were from the narrowest tertile, whereas few individuals were identified from the widest tertile, as hypothesized. Using tertile-specific averages as the reference group resulted in a more uniform distribution of individuals with lower Ct.Ar, as expected. This discordance in identifying individuals with lower Ct.Ar between the two methods is consistent with the variation in Ct.Ar among the robustness tertiles. Randomly sorting individuals into subgroups led to no difference in Ct.Ar among the randomized tertiles and resulted in almost perfect agreement for the two comparison methods. Thus, the traditional method using group averages as the reference for assessing when a child has low Ct.Ar failed to account for the variation in structure among children. Our findings have identified external bone size as a new source of bias that may exist within populations of females and males, underscoring the need to reframe how low bone mass is assessed in children.

The differences in Ct.Ar among the robustness tertiles reflects the complex adaptive nature of bone. Our findings have implications for bone health assessment by shining a light on the coordinate adjustment of multiple traits during growth, which results in bones that are sufficiently strong for daily activities but not overly heavy [33]. Because bone strength is related to the third power of bone width, the narrower bone phenotype has the potential to be extremely weak if the coordinate adjustment does not maximize Ct.Ar. To maximize strength, narrower bones adapt by having a proportionally thicker cortex and a larger Ct.Ar relative to Tt.Ar. Despite this adjustment, narrower metacarpals have a lower Ct.Ar on an absolute basis compared to wider metacarpals (Fig 3). At the other extreme, the wider bone phenotype has the external size to be strong but limits mass accumulation, otherwise the bone would be heavy. As such, wider bones tend to have a proportionally thinner cortex, but this is associated with higher Ct.Ar on an absolute basis. Lack of consideration of these structural differences, as shown in Tables 3 and 4, led to disparities in the assessment of low Ct.Ar. Thus, the coordinate adjustment of bone traits associated with the complex adaptive nature of bone means that comparing a child's bones to their external size peers is necessary to assess whether they have lower than expected bone mass.

Our research highlights the need for improved methods that lack bias in assessing pediatric bone health. Our prior work identified morphological measures of the metacarpal that lack a sex-specific bias when used to estimate strength [20], which will be important for intersex [34] and transgender individuals [35] who may be better supported clinically by having a predictor of bone strength without a sex-specific discrepancy [35]. Herein, we exposed external bone size as a source of bias that exists within populations. We propose that considering the different sets of traits arising from the complex adaptive nature of bone will help to minimize bias when assessing low bone mass. Pediatric bone health assessment is challenging due to the dynamic nature of bone growth [36,37]. Traditional methods for assessing bone health often fail to account for structural differences, leading to misdiagnosis [38]. Although it has yet to be determined whether narrower bones are over diagnosed with lower bone mass compared to wider bones in clinical settings, our data strongly advocate for adjusting bone mass for external size to prevent these inaccuracies and ensure fair assessments, ultimately improving the accuracy of pediatric bone health assessment [36].

**Table 4. Cohen's kappa coefficient showing the level of agreement between two methods used to identify females and males with lower Ct.Ar.** The two methods include comparing individual values to group averages versus robustness tertile-specific averages. Agreement outcomes are shown when lower Ct.Ar is defined as individuals having the lowest 50%, 33%, or 25% of Ct.Ar values for the group. The randomized-tertile data are shown for the lower 33% of Ct.Ar values for comparison to illustrate that the process of sorting individuals into tertiles did not affect the agreement. The midshaft structural data are the average of the 40, 50 and 60% sites.

**4a. Females**

| Robustness Tertile | Low bone mass thresholds | | | Randomized Tertile |
|---|---|---|---|---|
| | 50% | 33% | 25% | 33% |
| Narrow | 0.38 | 0.40 | 0.36 | **0.86** |
| Intermediate | 0.74 | **0.84** | **1.00** | **0.84** |
| Wide | 0.15 | 0.00 | 0.00 | **1.00** |

**4b. Males**

| Robustness Tertile | Low bone mass thresholds | | | Randomized Tertile |
|---|---|---|---|---|
| | 50% | 33% | 25% | 33% |
| Narrow | 0.40 | 0.47 | 0.48 | **0.85** |
| Intermediate | **1.00** | **0.85** | **0.85** | **1.00** |
| Wide | 0.56 | 0.53 | 0.49 | **0.88** |

**Interpretation of Cohen's kappa coefficient:**

0.00-0.20 (poor agreement), 0.21-0.40 (fair agreement), 0.41-0.60 (moderate agreement), 0.61-0.80 (substantial agreement), 0.81-1.00 (almost perfect agreement).

Our results suggest it may be important to recognize that lower than expected strength may arise in different ways [39] and that using a one-size-fits-all approach may not help to understand the underlying structural variation contributing to reduced bone strength. Our analysis shows lower Ct.Ar and strength could arise from having a well adapted narrow bone (i.e., narrow bone with the expected Ct.Ar) and/or a bone of varying external size with lower than expected mass accumulation. A greater proportion of females and males with narrow bones were identified as having lower Ct.Ar in the current study when Ct.Ar was compared to group averages. This bias has the potential to impose unnecessary testing and interventions on children, causing undue stress, medical expenses, and unwarranted concerns for them and their families. Having a narrower bone is within the normal range of variation in external bone size and it would be important to determine if individuals with narrower bones also have lower than expected mass accumulation. Our data emphasize the importance of comparing children with narrow bones to their structural peers (i.e., those with a similar external bone size) to make this assessment. Although narrower bones have a lower baseline strength [24,40], there is at least one advantage to having this structural phenotype. Recent research indicates that women and men with narrower femoral necks experience smaller reductions in bone mineral content (BMC) and greater area gains with age, suggesting this morphological phenotype may have lower strength early in life but demonstrate structural adaptations that maintain strength over time [41].

On the other hand, females and males with wider bones were underdiagnosed with lower Ct.Ar when the group average was used for comparison. Within the wide tertile, six males and five females were identified as having lower Ct.Ar, but fewer were diagnosed compared to the overall average. This underdiagnosis may result in missed opportunities for critical early interventions. Low mass in wider bones may be a concern later in life when age-related endosteal resorption occurs. The loss of bone mass with aging becomes more significant in wider bones with lower bone mass because loss from the endocortical surface occurs further from the centroid which would accelerate the strength decline [42]. Without timely treatment or lifestyle modifications, these individuals may face more significant bone health issues later in life. Thus, it is vital to monitor bone growth precisely, considering the full spectrum of variation in external bone size for different biomechanical reasons.

Our findings highlight the need to develop new diagnostic tools that offer a more tailored approach to bone mass assessment, whereby pediatricians would evaluate children within similar structural phenotypic groups to test for low bone mass. How treatment and intervention decisions benefit from the development of diagnostic tools that incorporate the variation in external bone size requires more research. Mouse models of narrow and wide phenotypes show different responses to OVX [43], bisphosphonate treatment [44], exercise [45], or a sclerostin neutralizing antibody treatment [46]. Different combinations of phenotype and treatment affected whole bone mechanical properties uniquely and the bone structure helped predict outcomes. External size and knowledge of different growth dynamics may provide clues to best treating individuals on a personalized basis [3,47]. Improving strength of a well adapted narrower bone (i.e., with expected Ct.Ar) may be challenging as the only option would be to target periosteal expansion since the marrow space is already minimized and new endosteal tissue would offer little mechanical benefit. Strategies for improving bone mass in wider bones may differ, given that Ma.Ar space is largely defined by osteoclastic resorption to expand the marrow space and to minimize mass.

This study has several limitations to address for a more comprehensive understanding of bone health assessment. First, the Bolton-Brush collection is limited to a specific demographic, focusing exclusively on ambulatory White females and males. This narrow demographic restricts the generalizability of our findings. Future research should include children from a wider range of backgrounds to understand how variation in bone structure affects low bone mass diagnoses across different racial/ethnic and ambulatory groups. Further, more robust data are needed to confirm that similar distributions also exist with known or suspected impairments to bone growth. The children enrolled in Bolton-Brush were healthy and we would not expect them to have a clinical diagnosis of low bone mass. Lower bone mass in the current study was created statistically using metacarpal Ct.Ar to illustrate that bones are built differently and that the reference population

matters when diagnosing low bone mass. Individuals with low bone mass in the general population would be expected to show more significant differences in Ct.Ar relative to their reference group than shown herein. Variation in bone age, which was not assessed in this study, would be expected to contribute to the variation in the structural measures, including Ct.Ar. Having had an independent measure of bone mass or strength, which was not available for this cohort, would have improved the generalizability of the results. Although cortical area has been shown to vary with external size for other long bone diaphyses of adult females and males [20,48] and bone mineral content has been shown to vary with the external size of the proximal femur of older females and males [41], additional studies are needed to confirm whether the association between the amount of bone and external bone size affects the identification of individuals with low bone mass at other anatomical sites. Ct.Ar was calculated assuming the metacarpal has a circular cross-section. We have reported that the cross-section could be better described as elliptical, with the long axis of the ellipse dependent on metacarpal width [20]. The extent to which the noncircular cross-sectional shape contributes to bias in identifying individuals with low bone mass has yet to be determined. Moreover, the analysis was limited to eight-year-old females and males. Expanding the age range could yield more robust data on how bone health assessment evolves through different stages of pediatric development. This broader age inclusion has the potential to significantly enhance the development of more accurate and age-appropriate diagnostic criteria. Finally, sorting females and males into tertiles was necessary to test the hypotheses and illustrate the impact of variation in external size on the diagnosis of low Ct.Ar. This subgroup method is not expected to be practical for the clinic. Alternative multi-variate approaches that assess bone mass as a continuous variable may be more practical for clinical use.

In conclusion, this study provided evidence that the variation in Ct.Ar relative to the external bone size of the metacarpal may be a source of bias when identifying individuals with low bone mass. Consideration of the different combinations of traits associated with external bone size may depend on the research or clinical question being asked. When assessing bone strength on an absolute basis, which may be needed to predict increased risk of insufficiency fractures [39,49], it may make sense to compare individuals to a population average. This comparison will identify individuals within a population having lower strength and likely result in many individuals with narrow bones being included in the low bone mass and low strength group. However, when assessing whether a perturbation, condition, or treatment affects the development of bone mass on an individualized basis, it is important to be specific about the reference population used to assess whether a person's Ct.Ar is lower than expected for their sex, age, and external bone size. Because metacarpal Ct.Ar varied with external bone size, comparing children to a reference group established using bone robustness provided a way to recognize individuals with lower Ct.Ar. Whereas this study did not attempt to establish absolute ranges for outer bone size sub-phenotyping, the data do suggest that recognizing patterns of structural variation that exist among individuals and that arise from the complex adaptive nature of bone may improve our ability to identify individuals developing a suboptimal skeleton. Efforts aimed at refining diagnostic measures may benefit from additional research identifying sources of bias that affect the identification of individuals with low bone mass.

## Acknowledgments

All authors declare they have no relationships/activities/interests that would pose real or perceived conflicts of interests with the material reported in this paper.

## Author contributions

**Conceptualization:** Karl J. Jepsen, Samantha Hertz, Finnegan Klein, Todd L. Bredbenner, Miranda Cosman.

**Data curation:** Karl J. Jepsen, Finnegan Klein.

**Formal analysis:** Karl J. Jepsen, Samantha Hertz, Finnegan Klein, Todd L. Bredbenner.

**Funding acquisition:** Karl J. Jepsen, Todd L. Bredbenner.

**Investigation:** Karl J. Jepsen, Samantha Hertz.

**Methodology:** Samantha Hertz, Todd L. Bredbenner.

**Project administration:** Karl J. Jepsen.

**Resources:** Karl J. Jepsen.

**Supervision:** Karl J. Jepsen, Miranda Cosman.

**Validation:** Karl J. Jepsen.

**Visualization:** Karl J. Jepsen, Finnegan Klein.

**Writing – original draft:** Karl J. Jepsen.

**Writing – review & editing:** Karl J. Jepsen, Samantha Hertz, Finnegan Klein, Todd L. Bredbenner, Miranda Cosman.

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
