## [Decision Letter · Decision Letter 0]

12 Aug 2025

PONE-D-25-37020

Identifying girls and boys with low bone mass from hand radiographs may be improved by considering the variation in metacarpal external size

PLOS ONE

Dear Dr. Jepsen,

Thank you for submitting your manuscript to PLOS ONE. After careful consideration, we feel that it has merit but does not fully meet PLOS ONE’s publication criteria as it currently stands. Therefore, we invite you to submit a revised version of the manuscript that addresses the points raised during the review process.

In addition to the Reviewers' comments, I note the following:

1. Please ensure the authors comply with PLOS One's data availability policy - data should be submitted to a public repository. 

2. While I appreciate the authors' expertise and previous work in this area, I find it somewhat disappointing that the authors are recommending development of yet another diagnostic tool for bone mass assessment. The bone research field has benefited from the introduction of tools such as pQCT & HR-pQCT, yet clinicians still rely mostly on DXA and this is the likely to remain the case. Is there a way for what is learned in this proof of concept study to be applied to other existing imaging tools beyond hand radiographs? 

3. A couple of minor formatting points: First, males/females is more appropriate when referring to biological sex, whereas boys/girls is used to refer to gender roles. Second, the abbreviation for cortical area can be used throughout the manuscript after it is first defined (which could be in the 2nd paragraph of the introduction on page 3). There are several places in the manuscript (other than at the start of a sentence) where cortical area is spelled out in full, but the abbreviation could be used instead (i.e., in the last sentence of the intro). 

4. Abstract, 4th sentence: Consider adding a definition for total area as was done for cortical area in this sentence. Currently, someone could read this as total and cortical area being measures of bone mass. 

5. Sample population: Does the current dataset represent all ~8 year olds in the Broadbent-Bolton Collection? Did the authors have to exclude any radiographs from the analysis. Also, at the top of page 5, the authors mention "the power needed to test the current hypothesis" but no information on a power / sample size calculation was provided in the manuscript. 

6. Bone structural measures: Please provide the units of measurement in the text. Please provide a reference for the statement regarding replacement of slenderness with robustness. 

7. Statistics: It would help if the authors could clarify how the Z-score thresholds were varied. 

8. Results: Relative cortical area is mentioned in the first paragraph, but this outcome wasn't defined in the methods. Similarly, BMI is presented in the tables, but is not mentioned in the text. 

We look forward to receiving your revised manuscript.

Kind regards,

Heather Macdonald, Ph.D

Academic Editor

PLOS ONE

Journal Requirements:

National Institute of Arthritis and Musculoskeletal and Skin Diseases of the National Institutes of Health (AR069620, AR068452;AR064244;AR082325)

All authors declare they have no relationships/activities/interests that would pose real or perceived conflicts of interests with the material reported in this paper. Research reported in this publication was supported in part by research grants from the National Institute of Arthritis and Musculoskeletal and Skin Diseases of the National Institutes of Health (KJJ: AR069620, AR068452; TLB: AR064244; KJJ & TLB: AR082325). The content is solely the responsibility of the authors and does not necessarily represent the official views of the National Institutes of Health.

National Institute of Arthritis and Musculoskeletal and Skin Diseases of the National Institutes of Health (AR069620, AR068452;AR064244;AR082325)

Reviewers' comments:

Reviewer's Responses to Questions

**Comments to the Author**

1. Is the manuscript technically sound, and do the data support the conclusions?

Reviewer #1: Yes

Reviewer #2: Yes

2. Has the statistical analysis been performed appropriately and rigorously?

Reviewer #1: Yes

Reviewer #2: Yes

3. Have the authors made all data underlying the findings in their manuscript fully available?

Reviewer #1: Yes

Reviewer #2: No

4. Is the manuscript presented in an intelligible fashion and written in standard English?

Reviewer #1: Yes

Reviewer #2: Yes

5. Review Comments to the Author

Reviewer #1: Identifying girls and boys with low bone mass from hand radiographs may be improved

by considering the variation in metacarpal external size

Thank you for the opportunity to review this manuscript. The research has been undertaken with rigour and adds to our understanding of bone structure in children. However, the manuscript would benefit from minor revision. My edits and comments are presented below.

Title

The title should be modified to sound less like a conclusion and more like a title.

Abstract

• Include whether left or right hand radiograph was used

Lay summary

• The first sentence refers to findings from a previous study – perhaps rephrase this sentence so that it reads as “background”.

• The second sentence should be rephrased for clarity “We showed in this study that this association…”

Introduction

• Line 4 – “The current diagnostic approach...” It is unclear what this phrase refers to.

• Paragraph 2 - A radiograph is the image obtained from x-ray exposure. Therefore, the term “radiograph” should be used instead of “x-ray” when referring to the image. This should be corrected throughout the manuscript.

• Paragraph 2 - A reference is required for “Another potential source of bias which may not be immediately obvious beyond sex and race/ethnicity is the variation in bone structure itself”

Materials and Methods

Sample population and data collection

• Page 4 The section below belongs in the Introduction of the manuscript where the background, rationale and objectives of the current study should be presented.

“Further, the current study builds on our prior work which established different external size dependent longitudinal trajectories for bone traits during growth (23, 24). Our prior work showed that the morphological traits of narrow bones grow along a different trajectory compared to wide. The current paper builds on our prior work by clarifying the differences in Ct.Ar among the narrow, intermediate, and wide subgroups, and then goes a step further to test if assessing low Ct.Ar can be improved by taking the variation in external size into consideration”

• “Our prior work showed that the morphological traits of narrow bones grow along a different trajectory compared to wide” – A reference is required

• “The current paper” should read “the current study”

• Specify whether the radiographs of the left or right hand

Bone Structural Measures

• Whilst the methodology has previously been published, details on the tools used to perform measurements should be included in this manuscript

• Page 5 – “Given that bigger people tend to have bigger bones” – specify what “bigger” means, and avoid the use of lay terms.

Analytical Methods

• Paragraph 1 – The following sentence should be rephrased or fragmented for clarity - “The primary hypothesis is that consideration of external size-dependent differences in bone structure will affect the identification of individuals with lower Ct.Ar compared to the standard method, which compares individuals to a group average.

Discussion

General comments

Given that radiographs of only White children were assessed, the manuscript would benefit from mentioning ethnic differences in metacarpal bone structure.

Specific comments

• Page 8 Paragraph 1 – “The primary outcome measure was metacarpal cortical area, which provides a dual measure of bone mass and strength…..” The metacarpal cortical area is not a direct measure of bone strength as stated, but is a proxy for bone strength.

• Page 9 – “Hand radiographs offer some advantages over conventional DXA methods by providing a more detailed view of bone structure….” Please clarify the highlighted phrase – what additional details do you get from a radiograph compared to DXA? Additionally, Reference 37 that has been used to support this sentence has been misinterpreted.

• Page 10 Paragraph 2 “Although narrower bones have a lower baseline strength (24, 40), there are some advantages to having this structural phenotype. Recent research indicates that women and men with narrower femoral necks….” Firstly, whilst the authors state that there are “some” advantages, only one is cited. Are there others? Secondly, given that later in the manuscript the authors mention the inability to generalize their results to other sites, it is odd that they have cited a study on the adult femoral neck to substantiate their comment.

• Page 12 paragraph 1 - “Associations between the development of the metacarpal and the development of other long bones was not investigated in this study. Consequently, we do not know if the results of the current study are generalizable to other anatomical sites”. Whilst the authors do not know whether their metacarpal results are applicable to other sites, perhaps they can discuss based on the known differences between metacarpals and other long bones, and from the results of previous studies that have made this comparison.

• Page 12 “In conclusion, this study provides evidence that the variation in bone structure arising from the complex adaptive nature of bone may be a source of bias when identifying individuals with low bone mass.” This study has not investigated any factors that influence bone size/structure. Therefore, I fail to understand the inclusion of the highlighted phrase.

Reviewer #2: Thank you for the opportunity to read and review this manuscript, “Identifying girls and boys with low bone mass from hand radiographs may be improved by considering the variation in metacarpal external size”. This research uses hand-wrist radiographs from a longitudinal study of known sex and age individuals to evaluate the impact that metacarpal robusticity has on bone mass. I found this manuscript interesting and well-written; it has a number of strengths, but there are a few areas (detailed in no particular order below) that may require some additional attention.

Some thoughts that I had while I was reading this manuscript:

I found the paper aims to be really interesting and insightful. While we frequently control for body size in our metacarpal analyses, this is often done to address sex-related size discrepancies. Based on the results of this paper, it seems clear that the bone robusticity (narrowness) does impact the cortical area, even after the element’s length is controlled for.

Perhaps my biggest question is in regards to the method and measurement justification. Specifically, the authors use the elliptical model method (EMM) to mathematically estimate the metacarpal cortical area from a single-plane radiograph. Typically, with a single plane X-ray (rather than a CT), I may choose a technique such as metacarpal radiogrammetry to measure the cortical thickness, which could then be corrected by the total width (thereby accounting for the bone’s robusticity/narrowness). Although it is a much simpler approach, metacarpal radiogrammetry has numerous clinical comparators, has been shown to correlate with bone mass, and does not mathematically extrapolate bone properties (as is necessary with EMM). While I do not disagree with the use of cortical area (there are many great reasons why it might be necessary!), I was curious: in this research, why is cortical area the necessary analytical parameter? Is this the standard in children’s bone mass analyses? Or is it because cortical area is specifically linked with a biomechanical attribute? I do not think it would be a large addition, but I would like to see the authors acknowledge/explain their use of the EMM over the more conventional metacarpal radiogrammetric approaches (i.e., why EMM / why not radiogrammetry).

I found Figure 2 quite visually effective. Nice addition.

Statistical reporting: I would encourage the authors to report their statistics in full, including the test result and degrees of freedom (when applicable), in addition to the p-value.

I found myself getting a little turned around with the difference between tertile and robustness tertile in the results section and in the figures/tables. For example, I understood that the robustness categories were also defined by tertiles – so interpreting the differences between Fig 3 B & C was a little difficult initially. I presume C is the randomized tertile? In text, for example, pg 7 para 2: “Relative cortical area differed among the tertiles for males but not for females.” – does this mean it differed for both randomized AND for robustness tertiles? Or just randomized? I would find a way to refer to these tertile groups in distinct ways in the text, tables, and figures. Perhaps once you introduce them as having been generated as tertiles, they might be called robustness tertiles vs. randomized tertiles? And these more descriptive categorical labels should be applied to the figures as well.

Pg 9, para 2 – this paragraph was really nicely discussed – I found it particularly helpful in better understanding the narrow vs. wide strength argument in relation to the author’s findings. Nice.

How would the authors propose these findings be applied in a clinical setting, in the absence of population standards? Based on these results, can a very general guideline be proposed to identify what might be considered low bone amounts as they relate to different metacarpal morphologies? If not, how should this work move forward so that children’s bone mass is being better quantified and categorized according to morphology?

Other minor edits:

- Pg 4, para 1: “For the metacarpal, cortical area (Ct.Ar) is a direct measure of the amount of bone (i.e., bone mass) for THE diaphysis…” (missing a word, could also perhaps read “in the diaphysis”, or “of the diaphysis”.

Overall, this manuscript offers valuable insight into factors, such as robusticity, that impact the ability to accurately assess and interpret low bone mass in metacarpals. I believe the findings reported in this study are important, and understanding the role of diaphyseal robusticity in quantifying cortical bone mass will be highly valuable for future assessments, potentially even in adults.

6. PLOS authors have the option to publish the peer review history of their article (what does this mean?). If published, this will include your full peer review and any attached files.

Reviewer #1: No

Reviewer #2: No

---

## [Author Response · Author response to Decision Letter 1]

15 Sep 2025

PONE-D-25-37020

Identifying girls and boys with low bone mass from hand radiographs may be improved by considering the variation in metacarpal external size

Response: We thank the Associate Editor and both reviewers for their enthusiasm of our paper and their constructive comments. We responded positively to all comments and feel the paper has been substantially improved. We look forward to the review of our revision and the opportunity to address any remaining questions.

EDITOR

In addition to the Reviewers' comments, I note the following:

1. Please ensure the authors comply with PLOS One's data availability policy - data should be submitted to a public repository.

Response: The measured and calculated metacarpal data used in this study are being uploaded to a publicly accessible resource. Please note that this data upload has just been initiated and will be complete within a few days. The doi for this data is: https://doi.org/10.7302/47ej-xb63

Access to the hand radiographs is publicly available but only through Bolton-Brush Growth Study Center at Case Western Reserve University; https://case.edu/dental/departments-programs/bolton-brush-growth-study-center

2. While I appreciate the authors' expertise and previous work in this area, I find it somewhat disappointing that the authors are recommending development of yet another diagnostic tool for bone mass assessment. The bone research field has benefited from the introduction of tools such as pQCT & HR-pQCT, yet clinicians still rely mostly on DXA and this is the likely to remain the case. Is there a way for what is learned in this proof of concept study to be applied to other existing imaging tools beyond hand radiographs?

Response: We would like to address the Associate Editor's comments with two points. First, we do not feel the current paper is recommending 'yet another' diagnostic tool for bone mass assessment; in fact, we are proposing a refinement of existing assessments to reduce bias within populations which is an area of research I believe many would feel is critically important. Further, the X-ray technology used to generate hand radiographs is more prevalent and thus more accessible than pQCT, HRpQCT, and DXA, which are expensive; as such, beginning this analysis using hand radiographs is highly appropriate for democratizing access to a diagnostic tool that minimizes bias. Second, we reframed the focus of the study to be less about advocating for use of hand radiographs compared to other systems to one where we are advocating adjustments for measures like external size which may be a source of bias. How the external size dependent differences in bone mass are observable in the various technologies (pQCT, HRpQCT, DXA) has yet to be determined. Given the results of this study, we would recommend that additional research is needed to study sources of bias in diagnosing low bone mass such as that reported herein.

Changes in the revision: As noted in the response to reviewer 1, we removed the section of the Discussion (page 9) that compares the advantages and limitations of different imaging technologies.

3. A couple of minor formatting points: First, males/females is more appropriate when referring to biological sex, whereas boys/girls is used to refer to gender roles. Second, the abbreviation for cortical area can be used throughout the manuscript after it is first defined (which could be in the 2nd paragraph of the introduction on page 3). There are several places in the manuscript (other than at the start of a sentence) where cortical area is spelled out in full, but the abbreviation could be used instead (i.e., in the last sentence of the intro).

Response: We replaced girls/boys with females/males throughout the paper and used Ct.Ar and Tt.Ar abbreviations after the first call out.

4. Abstract, 4th sentence: Consider adding a definition for total area as was done for cortical area in this sentence. Currently, someone could read this as total and cortical area being measures of bone mass.

Response: This sentence was modified as follows: "Total area (Tt.Ar), a measure of the area enclosed by the periosteal surface, and cortical area (Ct.Ar), a measure of bone mass, were calculated assuming a circular cross-section."

5. Sample population: Does the current dataset represent all ~8 year olds in the Broadbent-Bolton Collection? Did the authors have to exclude any radiographs from the analysis. Also, at the top of page 5, the authors mention "the power needed to test the current hypothesis" but no information on a power / sample size calculation was provided in the manuscript.

Response: We updated the manuscript to indicate that the radiographs were obtained for a random sampling of children enrolled in the Bolton-Brush study. page 4, "The hand radiographs were purchased from the Broadbent-Bolton Collection at the Bolton-Brush Growth Study Center in 2007 and represent a randomly selected subgroup of those enrolled in this study."

We corrected the missing power analysis by including the following text in the Methods section, page 5: "The combined dataset allowed us to test for an effect size as low as 0.45 when comparing the tertiles (3 groups) with 80% power and a significance level of 0.05. This study was powered to test for significant differences in Ct.Ar among tertiles, which showed an average effect size of 1 for females and males."

6. Bone structural measures: Please provide the units of measurement in the text. Please provide a reference for the statement regarding replacement of slenderness with robustness.

Response: We removed this sentence to reduce confusion as the prior two papers reporting bone morphology from the Bolton-Brush used the term 'robustness' rather than slenderness. Our inclusion of this sentence refers to other papers that are not germane to the current study and so there was no need for this clarification.

7. Statistics: It would help if the authors could clarify how the Z-score thresholds were varied.

Response: We added the following text to the Results section (page 8) to clarify how the Z-score thresholds were varied: "For females, the Z-score thresholds were -0.51 and -0.40 so 33% of individuals were identified with low bone mass when compared to the group average and tertile-specific average, respectively. For males, the Z-score thresholds were -0.70 and -0.36 so 33% of individuals were identified with low bone mass when compared to the group average and tertile-specific average, respectively."

8. Results: Relative cortical area is mentioned in the first paragraph, but this outcome wasn't defined in the methods. Similarly, BMI is presented in the tables but is not mentioned in the text.

Response: We added new text to the Methods section to define BMI (page 5: "Body Mass Index (BMI) was calculated as body weight divided by height squared.") and relative cortical area (page 6: "Relative cortical area (RCA) was calculated as Ct.Ar / Tt.Ar.")

Reviewer #1: Thank you for the opportunity to review this manuscript. The research has been undertaken with rigour and adds to our understanding of bone structure in children. However, the manuscript would benefit from minor revision. My edits and comments are presented below.

Response: We thank the reviewer for the kind words and constructive comments.

1. Title: The title should be modified to sound less like a conclusion and more like a title.

Response: The new title reads, "Impact of external size on the identification of low bone mass in children"

2. Abstract

• Include whether left or right hand radiograph was used

Response: Hand radiographs were obtained for the nondominant hand. This designation is now included in the Abstract (as suggested) and in the Methods (page 4).

3. Lay summary

• The first sentence refers to findings from a previous study – perhaps rephrase this sentence so that it reads as “background”.

Response: The first sentence now reads, "In prior work, we reported that bone mass varied with outer bone size."

• The second sentence should be rephrased for clarity “We showed in this study that this association…”

Response: The second sentence now reads, "In the current study, we showed that this association affected the identification of female and male children with low metacarpal cortical area."

4. Introduction

• Line 4 – “The current diagnostic approach...” It is unclear what this phrase refers to.

Response: The sentence was revised to provide clarity regarding the diagnostic approach we are referring to: "The current approach to assessing bone strength is tailored for rare diseases and related conditions but is neither beneficial for the general population nor effective in establishing comparative norms."

• Paragraph 2 - A radiograph is the image obtained from x-ray exposure. Therefore, the term “radiograph” should be used instead of “x-ray” when referring to the image. This should be corrected throughout the manuscript.

Response: Thank you for pointing this out. We replaced X-ray with radiograph throughout the manuscript, except when used in the titles of cited papers.

• Paragraph 2 - A reference is required for “Another potential source of bias which may not be immediately obvious beyond sex and race/ethnicity is the variation in bone structure itself”

Response: We cited our prior study examining how external bone size affects the association between metacarpal structure and strength.

E. M. R. Bigelow, R. W. Goulet, A. Ciarelli, S. H. Schlecht, D. H. Kohn, T. L. Bredbenner, et al., Sex and external size specific limitations in assessing bone health from adult hand radiographs, JBMR Plus 2022 Vol. 6 Issue 8 Pages e10653

5. Materials and Methods

Sample population and data collection

• Page 4 The section below belongs in the Introduction of the manuscript where the background, rationale and objectives of the current study should be presented.

“Further, the current study builds on our prior work which established different external size dependent longitudinal trajectories for bone traits during growth (23, 24). Our prior work showed that the morphological traits of narrow bones grow along a different trajectory compared to wide. The current paper builds on our prior work by clarifying the differences in Ct.Ar among the narrow, intermediate, and wide subgroups, and then goes a step further to test if assessing low Ct.Ar can be improved by taking the variation in external size into consideration”

Response: Thank you for this suggestion. We moved this text to the Introduction (page 4), which now connects our overall premise to our prior longitudinal study more clearly.

• “Our prior work showed that the morphological traits of narrow bones grow along a different trajectory compared to wide” – A reference is required

Response: Our prior two studies (Pandey et al, JBMR 2009; Bhola et al, Bone 2013) are now included in this sentence.

• “The current paper” should read “the current study”

Response: The 'current paper' was replaced with 'current study.'

• Specify whether the radiographs of the left or right hand

Response: Hand radiographs were taken of the nondominant hand. This is now included in the Abstract and Methods sections.

6. Bone Structural Measures

• Whilst the methodology has previously been published, details on the tools used to perform measurements should be included in this manuscript

Response: There were not many more details to add regarding the measurement protocols except to clarify that the measures were conducted manually using the software and that the coefficient of variation for repeat point to point measures was 1.61%. This information is now included in the revised Methods section (page 5).

• Page 5 – “Given that bigger people tend to have bigger bones” – specify what “bigger” means, and avoid the use of lay terms.

Response: The reviewer asks a good question and our apologies for the laid-back terminology. We removed this parenthetical phrase to avoid using lay terms and to just state that Tt.Ar was adjusted for bone length to take the inter-individual variation in body size into consideration.

7. Analytical Methods

• Paragraph 1 – The following sentence should be rephrased or fragmented for clarity - “The primary hypothesis is that consideration of external size-dependent differences in bone structure will affect the identification of individuals with lower Ct.Ar compared to the standard method, which compares individuals to a group average.

Response: The sentence was rewritten as follows: "We tested two hypotheses in the current study. First, we tested whether Ct.Ar differed among subgroups sorted based on external bone size. Second, we tested whether identifying individuals with lower bone mass would be improved when comparing Ct.Ar of individuals to the average of their external size subgroups versus comparing Ct.Ar of individuals to the overall cohort average."

8. Discussion

General comments

Given that radiographs of only White children were assessed, the manuscript would benefit from mentioning ethnic differences in metacarpal bone structure.

Response: The paragraph describing the limitations of the paper (page 11) includes a statement regarding the need to conduct a similar study in other databases having more diverse representation of children based on racial, ethnic, and ambulatory backgrounds.

"This study has several limitations to address for a more comprehensive understanding of bone health assessment. First, the Bolton-Brush collection is limited to a specific demographic, focusing exclusively on ambulatory White girls and boys. This narrow demographic restricts the generalizability of our findings. Future research should include children from a wider range of backgrounds to understand how variation in bone structure affects low bone mass diagnoses across different racial/ethnic and ambulatory groups."

9. Specific comments

• Page 8 Paragraph 1 – “The primary outcome measure was metacarpal cortical area, which provides a dual measure of bone mass and strength…..” The metacarpal cortical area is not a direct measure of bone strength as stated, but is a proxy for bone strength.

Response: We rephrased this statement to read - "The primary outcome measure was metacarpal cortical area, which provides a measure of bone mass and a proxy for whole bone strength that lacks a sex-specific bias (21)."

• Page 9 – “Hand radiographs offer some advantages over conventional DXA methods by providing a more detailed view of bone structure….” Please clarify the highlighted phrase – what additional details do you get from a radiograph compared to DXA? Additionally, Reference 37 that has been used to support this sentence has been misinterpreted.

Response: We removed this section of the Discussion as a comparison of the advantages and limitations of different imaging technologies is not germane to the study outcomes.

• Page 10 Paragraph 2 “Although narrower bones have a lower baseline strength (24, 40), there are some advantages to having this structural phenotype. Recent research indicates that women and men with narrower femoral necks….” Firstly, whilst the authors state that there are “some” advantages, only one is cited. Are there others? Secondly, given that later in the manuscript the authors mention the inability to generalize their results to other sites, it is odd that they have cited a study on the adult femoral neck to substantiate their comment.

Response: There may be other advantages but we only cite the one; as such the sentence was rewritten as follows: "Although narrower bones have a lower baseline strength (24, 37), there is at least one advantage to having this structural phenotype."

The inconsistency noted by the reviewer regarding whether the outcomes of our study of the metacarpal would be observed for other long bones is addressed in your next question.

• Page 12 paragraph 1 - “Associations between the development of the metacarpal and the development of other long bones was not investigated in this study. Consequently, we do not know if the results of the current study are generalizable to other anatomica

---

## [Decision Letter · Decision Letter 1]

19 Oct 2025

PONE-D-25-37020R1Impact of external size on the identification of low bone mass in childrenPLOS ONE

Dear Dr. Jepsen,

Thank you for addressing the Reviewers' comments. Upon second review, there are additional comments to consider. Therefore, we invite you to submit a second revision that addresses the points raised by the Reviewers.

We look forward to receiving your revised manuscript.

Kind regards,

Heather Macdonald, Ph.D

Academic Editor

PLOS ONE

Journal Requirements:

Reviewers' comments:

Reviewer's Responses to Questions

**Comments to the Author**

1. If the authors have adequately addressed your comments raised in a previous round of review and you feel that this manuscript is now acceptable for publication, you may indicate that here to bypass the “Comments to the Author” section, enter your conflict of interest statement in the “Confidential to Editor” section, and submit your "Accept" recommendation.

Reviewer #1: (No Response)

Reviewer #2: (No Response)

2. Is the manuscript technically sound, and do the data support the conclusions?

Reviewer #1: Yes

Reviewer #2: Yes

3. Has the statistical analysis been performed appropriately and rigorously?

Reviewer #1: Yes

Reviewer #2: Yes

4. Have the authors made all data underlying the findings in their manuscript fully available?

Reviewer #1: (No Response)

Reviewer #2: Yes

5. Is the manuscript presented in an intelligible fashion and written in standard English?

Reviewer #1: Yes

Reviewer #2: Yes

6. Review Comments to the Author

Reviewer #1: Impact of external size on the identification of low bone mass in children

The reviewer thanks the authors for addressing the comments. Whilst mostly this has done been satisfactorily, there are still points which require clarification.

Title

Whilst the revised title now reads less like a conclusion, it still needs improvement! What does “external size” refer to?

Similarly, the revised sentence on p5 needs to clarify ‘external size”

Abstract

The issue of hand dominance remains unclear.

The authors were asked to specify whether left or right hand radiographs were used. The authors instead now mention use of the radiograph of the non-dominant hand. Does this mean that there was a record of whether an individual was sinistral or dextral, and that a combination of left and right hand radiographs was used? Or did they assume that all individuals were right-hand dominant?

Kindly address this in Methods as well

Introduction

Page 3 - “A review of morphological measures derived from hand radiographs (20) revealed that many widely used measures correlated with experimentally determined strength but with a sex-specific discrepancy (21).” – reference 20 is inappropriately placed. Are both refs 20 and 21 required? Citing an empirical study rather than a review is preferred.

Page 4 - “Narrower bones have proportionally thicker cortices but lower cortical areaCt.Ar (i.e., mass) compared to wider bones, but adjustments for this morphological variation have not been considered clinically or for research.” – the highlighted portion is inaccurate. I suggest the authors review the longitudinal studies on metacarpal bones undertaken in South African children.

Refs 22 and 23 were studies done on the same data set as the current study i.e. white individuals. Therefore, I fail to see how these 2 refs were used to cite racial/ethnic differences in bone.

Methods

Page 6

“Body Mass Index (BMI) was calculated as body weight divided by height squared.” – please provide units of measurement.

Were the radiographs digitised or hard copies? In the context of my question, please clarify how measurements were then done “manually”.

“Manual point-to-point measurements of outer and inner bone widths had an average coefficient of variation of 1.61%.” This appears to be a citation from ref 23 where data for only 29 boys and girls were used. Given the differences in sample sizes, it seems coincidental that the coefficient of variation in this study is identical to that of the previous study (ref 23)? Kindly explain.

Reviewer #2: Thank you for the opportunity to read and review the revised version of this manuscript, “Impact of external size on the identification of low bone mass in children”. This research uses hand-wrist radiographs from a longitudinal study of known sex and age individuals to evaluate the impact that metacarpal robusticity has on bone mass. I thought the authors did a good job of integrating and applying the reviewer feedback. I am happy with the revisions, but they have raised a few additional comments (detailed in no particular order below) that may require some additional attention.

Some thoughts that I had while I was reading this manuscript:

In the response to reviewers, the authors reiterate how important it was that they could pool sexes for these analyses. Initially, I found this a little unclear as to why – especially because on pg 4 the authors state the reference groups are often split by sex and age, and the sex differences mentioned for MCI measures made me worry that sex was indeed a confounding factor that should be controlled for. *However*, nearer the end of the paper, the authors bring up an excellent point about how a non-sex specific approach is more inclusive in cases of sex- and gender-diversity. This is an excellent point, and one that could perhaps be emphasized in the earlier justifications for why this study aimed to derive the sample they did (i.e., both sexes pooled).

Pg 7, Results para 1: An ANOVA p-value is reported without the associated F and df. These are likely in the table, but if the p-value is presented in text, perhaps the authors could also consider in-text presentation?

Statistics presentation: I would encourage the authors to adhere to other, more common or accepted guidelines for how to present df and F statistics alongside p-values. For ANOVA, this may look like (a completely fictional example): (F(2)=9.073, p=.031). The ‘2’ in brackets after the F is the df. This reporting makes for a cleaner/tidier presentation of the stats in a way that matches convention in many studies. I would also encourage the tabulated statistics to be presented the same.

The final concluding sentence: I understand this is a new sentence addition, but I am not sure that it helps to clearly summarize your paper, or its findings and implications. Specifically, I am not convinced that we actually need new approaches, or that this paper supports that need specifically. We might, however, need measured ranges that can advise clinicians when to further investigate an individual as ‘at risk’ for low bone mass; I understand that these absolute ranges are not something that the authors can provide within the scope of this study. Perhaps instead, these final sentences may reiterate that the study did not endeavor to provide absolute narrowness/wideness values/ranges of concern for low bone mass, but rather to establish that bone width can misdirect, and therefore cannot be accurately used to identify loss conclusively. Based on the results of this research, future studies and clinical practice should be aware that even wide bones may still have strength and mass concerns/issues.

7. PLOS authors have the option to publish the peer review history of their article (what does this mean?). If published, this will include your full peer review and any attached files.

Reviewer #1: No

Reviewer #2: No

---

## [Author Response · Author response to Decision Letter 2]

21 Oct 2025

Comments to the Author

Reviewer #1: Impact of external size on the identification of low bone mass in children

The reviewer thanks the authors for addressing the comments. Whilst mostly this has done been satisfactorily, there are still points which require clarification.

Title

Whilst the revised title now reads less like a conclusion, it still needs improvement! What does “external size” refer to? Similarly, the revised sentence on p5 needs to clarify ‘external size”

Response: External size was replaced with "outer width of the metacarpal diaphysis" in the title. External bone size was defined in the Abstract (page 2) and when it was first used in the Introduction (page 3).

New text (page 2; Abstract): "Prior work showed that bone mass varied with external bone size, a measure of the outer bone width."

New text (page 4; Introduction): "External bone size represents any measure describing the outer size of the bone, including outer width and total cross-sectional area."

Abstract

The issue of hand dominance remains unclear.

The authors were asked to specify whether left or right hand radiographs were used. The authors instead now mention use of the radiograph of the non-dominant hand. Does this mean that there was a record of whether an individual was sinistral or dextral, and that a combination of left and right hand radiographs was used? Or did they assume that all individuals were right-hand dominant?

Kindly address this in Methods as well

Response: We would assume the study records included an assessment of hand dominance, but we did not have access to this information. The majority of the radiographs examined were of the left hand, which is consistent with the statistic that approximately 90% of individuals are right hand dominant.

New text (page 5, Methods): "The majority of the radiographs examined were of the left hand, which is consistent with the general statistic that approximately 90% of individuals are right hand dominant."

Introduction

Page 3 - “A review of morphological measures derived from hand radiographs (20) revealed that many widely used measures correlated with experimentally determined strength but with a sex-specific discrepancy (21).” – reference 20 is inappropriately placed. Are both refs 20 and 21 required? Citing an empirical study rather than a review is preferred.

Response: We removed the reference to the review paper and only used the empirical study when referring to prior work showing a sex-specific discrepancy between morphological measures and bone strength.

Page 4 - “Narrower bones have proportionally thicker cortices but lower cortical area, Ct.Ar (i.e., mass) compared to wider bones, but adjustments for this morphological variation have not been considered clinically or for research.” – the highlighted portion is inaccurate. I suggest the authors review the longitudinal studies on metacarpal bones undertaken in South African children.

Response: We thank the reviewer for pointing out this error in referencing prior work. We included two papers reporting on longitudinal bone changes in South African children as examples of studies which took into consideration the associations between external bone size and the amount of bone.

New text (page 4, Introduction): "Narrower bones have proportionally thicker cortices but lower Ct.Ar (i.e., mass) compared to wider bones {Magan, 2017 #2112;Magan, 2019 #1956}."

Refs 22 and 23 were studies done on the same data set as the current study i.e. white individuals. Therefore, I fail to see how these 2 refs were used to cite racial/ethnic differences in bone.

Response: Thank you for pointing this out. Although the references were intended to point out the range in external sizes and not work conducted on multiple races/ethnicities, we see now how this was an error. We adjusted the text to remove the reference to races/ethnicities and to simply reference prior work examining the variation in external bone size independent of body stature.

New text (page 4; Introduction): "External bone size varies within female and male populations, ranging from narrow to wide phenotypes, even after adjusting for body stature (21, 22)."

Methods

Page 6. “Body Mass Index (BMI) was calculated as body weight divided by height squared.” – please provide units of measurement.

Response: Units are now included in the revised manuscript.

New text (page 6; Methods): "Body Mass Index (BMI) was calculated as body weight (kilograms) divided by height squared (meters2).”

Were the radiographs digitised or hard copies? In the context of my question, please clarify how measurements were then done “manually”.

Response: The measurements were determined from digitized copies of the original radiographs. The text was updated to reflect this detail. Measurements were done manually in the sense that the student identified the edges of the outer and inner surfaces of the diaphysis and did not use an edge detection algorithm.

New text (page 5, Methods): "Digitized hand radiographs were purchased from the Broadbent-Bolton Collection at the Bolton-Brush Growth Study Center in 2007 and represent a randomly selected subgroup of those enrolled in this study."

New text (page 6, Methods): "Manual point-to-point measurements refer to the identification of the edges of the outer and inner surfaces of the diaphysis by an individual and not by an edge detection algorithm."

“Manual point-to-point measurements of outer and inner bone widths had an average coefficient of variation of 1.61%.” This appears to be a citation from ref 23 where data for only 29 boys and girls were used. Given the differences in sample sizes, it seems coincidental that the coefficient of variation in this study is identical to that of the previous study (ref 23)? Kindly explain.

Response: The repeatability study was conducted at the time the measurements were taken using a random collection of hand radiographs (n=5) with 10 repeat measurements each for the outer and inner widths. The coefficient of variation was averaged for the outer and inner width measures. The Methods section was updated to include this information.

New text (page 6, Methods): "A repeatability study was conducted at the time the measurements were taken using a random collection of five hand radiographs with 10 repeat measurements for each of the outer and inner widths."

Reviewer #2: Thank you for the opportunity to read and review the revised version of this manuscript, “Impact of external size on the identification of low bone mass in children”. This research uses hand-wrist radiographs from a longitudinal study of known sex and age individuals to evaluate the impact that metacarpal robusticity has on bone mass. I thought the authors did a good job of integrating and applying the reviewer feedback. I am happy with the revisions, but they have raised a few additional comments (detailed in no particular order below) that may require some additional attention.

Some thoughts that I had while I was reading this manuscript:

In the response to reviewers, the authors reiterate how important it was that they could pool sexes for these analyses. Initially, I found this a little unclear as to why – especially because on pg 4 the authors state the reference groups are often split by sex and age, and the sex differences mentioned for MCI measures made me worry that sex was indeed a confounding factor that should be controlled for. *However*, nearer the end of the paper, the authors bring up an excellent point about how a non-sex specific approach is more inclusive in cases of sex- and gender-diversity. This is an excellent point, and one that could perhaps be emphasized in the earlier justifications for why this study aimed to derive the sample they did (i.e., both sexes pooled).

Response: We apologize for the confusion. The male and female data were not pooled but analyzed separately. We reviewed the Methods section and made some tweaks to the text to ensure this point was clear.

Revised text (page 6, Methods): "The female and male datasets allowed us to test for an effect size as low as 0.45 when comparing the tertiles (3 groups) with 80% power and a significance level of 0.05." "This study was powered to test for significant differences in Ct.Ar among tertiles, which showed an average effect size of 1 for females and males when analyzed separately."

Revised text (page 7, Analytical Methods): "We tested two hypotheses in the current study with females and males being analyzed separately."

Revised text (page 10, Discussion): "Hand radiographs from eight-year-old females and males were examined and analyzed separately as a proof-of-concept to test whether metacarpal external size affected the identification of children with lower Ct.Ar."

Pg 7, Results para 1: An ANOVA p-value is reported without the associated F and df. These are likely in the table, but if the p-value is presented in text, perhaps the authors could also consider in-text presentation?

Response: The Results section now includes ANOVA details for body weight, height, and metacarpal length.

Revised text (page 9, Results): "Weight (Females: F(2)=0.16, p=0.849; Males: F(2)=1.00, p=0.376; ANOVA), height (Females: F(2)=1.51, p=0.234; Males: F(2)=0.36, p=0.698; ANOVA), and metacarpal length (Females: F(2)=0.40, p=0.673; Males: F(2)=0.002, p=0.998; ANOVA) did not differ among the robustness tertiles for either sex (Table 2), indicating that sorting females and males based on second metacarpal robustness reflected differences in outer bone size but not body stature or bone length."

Statistics presentation: I would encourage the authors to adhere to other, more common or accepted guidelines for how to present df and F statistics alongside p-values. For ANOVA, this may look like (a completely fictional example): (F(2)=9.073, p=.031). The ‘2’ in brackets after the F is the df. This reporting makes for a cleaner/tidier presentation of the stats in a way that matches convention in many studies. I would also encourage the tabulated statistics to be presented the same.

Response: The revision now includes the recommended reporting format for ANOVA throughout the Results section and Table 2.

The final concluding sentence: I understand this is a new sentence addition, but I am not sure that it helps to clearly summarize your paper, or its findings and implications. Specifically, I am not convinced that we actually need new approaches, or that this paper supports that need specifically. We might, however, need measured ranges that can advise clinicians when to further investigate an individual as ‘at risk’ for low bone mass; I understand that these absolute ranges are not something that the authors can provide within the scope of this study. Perhaps instead, these final sentences may reiterate that the study did not endeavor to provide absolute narrowness/wideness values/ranges of concern for low bone mass, but rather to establish that bone width can misdirect, and therefore cannot be accurately used to identify loss conclusively. Based on the results of this research, future studies and clinical practice should be aware that even wide bones may still have strength and mass concerns/issues.

Response: The concluding paragraph was modified, including removing the last sentence.

Revised text (page 14/15, Discussion): "Whereas this study did not attempt to establish absolute ranges for outer bone size sub-phenotyping, the data do suggest that recognizing patterns of structural variation that exist among individuals and that arise from the complex adaptive nature of bone may improve our ability to identify individuals developing a suboptimal skeleton. Efforts aimed at refining diagnostic measures may benefit from additional research identifying sources of bias that affect the identification of individuals with low bone mass."

---

## [Decision Letter · Decision Letter 2]

26 Oct 2025

Impact of outer width of the metacarpal diaphysis on the identification of low bone mass in children

PONE-D-25-37020R2

Dear Dr. Jepsen,

We’re pleased to inform you that your manuscript has been judged scientifically suitable for publication and will be formally accepted for publication once it meets all outstanding technical requirements. I also note a few minor edits, below. 

Kind regards,

Heather Macdonald, Ph.D

Academic Editor

PLOS ONE

Additional Editor Comments (optional):

Thank you for addressing the reviewers' comments. I have a couple of minor suggestions/edits:

- If the word count allows, consider mentioning in the abstract that the midshaft structural data are the average of measures taken at the 40, 50 and 60% sites. I would also suggest noting this in footnotes for the tables (or put midshaft in the table captions), as currently, it isn't clear in the tables at what site the structural measures are from.

- In Figures 1 and 3, change CtAr to Ct.Ar as per the rest of the paper.

- In Figure 3, please indicate in the figure caption what the error bars represent. Please also consider including the exact p-values for those tertile comparisons that are statistically significant. 

Reviewers' comments:

Reviewer's Responses to Questions

**Comments to the Author**

1. If the authors have adequately addressed your comments raised in a previous round of review and you feel that this manuscript is now acceptable for publication, you may indicate that here to bypass the “Comments to the Author” section, enter your conflict of interest statement in the “Confidential to Editor” section, and submit your "Accept" recommendation.

Reviewer #1: All comments have been addressed

2. Is the manuscript technically sound, and do the data support the conclusions?

Reviewer #1: Yes

3. Has the statistical analysis been performed appropriately and rigorously?

Reviewer #1: Yes

4. Have the authors made all data underlying the findings in their manuscript fully available?

Reviewer #1: Yes

5. Is the manuscript presented in an intelligible fashion and written in standard English?

Reviewer #1: Yes

6. Review Comments to the Author

Reviewer #1: Thank you for addressing all my comments. I suggest that the manuscript be accepted for publication.

7. PLOS authors have the option to publish the peer review history of their article (what does this mean?). If published, this will include your full peer review and any attached files.

Reviewer #1: No

---

## [Editor Report · Acceptance letter]

PONE-D-25-37020R2

PLOS ONE

Dear Dr. Jepsen,

I'm pleased to inform you that your manuscript has been deemed suitable for publication in PLOS ONE. Congratulations! Your manuscript is now being handed over to our production team.

Kind regards,

on behalf of

Dr. Heather Macdonald

Academic Editor

PLOS ONE